# Beyond Neural Incompatibility: Easing Cross-Scale Knowledge Transfer in Large Language Models through Latent Semantic Alignment

## Abstract

Large Language Models (LLMs) encode vast amounts of knowledge in their massive parameters, which is accessible to locate, trace, and analyze. Despite advances in neural interpretability, it is still not clear how to transfer knowledge in a fine-grained manner, namely parametric knowledge transfer (PKT). A key problem is enabling effective and efficient knowledge transfer across LLMs of different scales, which is essential for achieving greater flexibility and broader applicability in transferring knowledge between LLMs. Due to neural incompatibility, referring to the architectural and parametric differences between LLMs of varying scales, existing methods that directly reuse layer parameters are severely limited. In this paper, we identify the semantic alignment in latent space as the fundamental prerequisite for LLM cross-scale knowledge transfer. Instead of directly using the layer parameters, our approach takes activations as the medium of layer-wise knowledge transfer. Leveraging the semantics in latent space, our approach is simple and outperforms prior work, better aligning model behaviors across varying scales. Evaluations on four benchmarks demonstrate the efficacy of our method. Further analysis reveals the key factors easing cross-scale knowledge transfer and provides insights into the nature of latent semantic alignment.

## 1 Introduction

Language is the channel that lets humans and today's language models communicate, yet it throws away much of the fine detail that lives inside a model. When we teach a smaller model using instructions, explanations, or distilled datasets, we compress the teacher's rich internal signals into text and lose structure that matters for behavior. A better way to transfer knowledge would move internal states directly, similar in spirit to the idea of brainwave communication where the sender shares what it is thinking rather than what it can say. Large language models make this idea practical because their parameters and hidden states are accessible. Prior work shows that we can analyze these internals, find where knowledge lives, and measure how specific parts influence predictions using attribution methods and information flow tools (Kokhlikyan et al., 2020; Yu & Ananiadou, 2024; Ferrando & Voita, 2024; Chen et al., 2025). This sets up our motivation for better knowledge transfer, where a student should be able to receive those signals directly from a teacher without going through text. It promises less loss, lower cost, and more truthful transfer.

We study this idea under parametric knowledge transfer. The goal is to move internal knowledge from a larger teacher to a smaller student so that the student acts more like the teacher. Prior work explores two routes in parameter space. Seeking extracts teacher parameters with sensitivity measures, injects them into the student through a LoRA initialization, and then relies on post alignment fine tuning (Zhong et al., 2024). LaTen aligns parameter spaces before injection using a light mapping to reduce the cost of later training (Tan et al., 2025). Both show that knowledge transfer is possible, but also report instability when the models differ in module design and parameter values, a gap described as *neural incompatibility* (Tan et al., 2025). In our analogy above, the "brainwave communication" corresponds to sharing layer outputs, not sharing layer parameters. We therefore treat activations as the medium of transfer and align semantics first, before any parameter updates.

We propose SEMALIGN, a semantics-first method for parametric knowledge transfer that uses layer outputs as the transfer signal. SemAlign consists of three steps: First, we run layer attribution on the teacher to locate layers that carry task relevant signal and we pair them with compatible layers in the student. This follows evidence that neuron and concept relations are many to many and that robust layer selection matters (Yu & Ananiadou, 2024; Chen et al., 2025). Second, we align latent semantics for each paired layer. We decompose the teacher hidden states into semantic components in the teacher space and recombine them as supervisory hidden states in the student space. This treats aligned activations as supervision and follows results showing that shaping hidden states preserves meaning and supports stable adaptation (Gu et al., 2024; 2025; Kong et al., 2024). Third, we steer the student by optimizing the paired layers so that, on the same inputs, their outputs approach the aligned supervisory hidden states. In short, we align how layers behave rather than how weights look, which reduces neural incompatibility while keeping the procedure simple and efficient.

We evaluate SemAlign under the same setup as the prior work (Tan et al., 2025). We use four standard benchmarks on professional knowledge, mathematical reasoning and code generation. The experiments are conducted with Llama 2 models (Touvron et al., 2023), performing task related parametric knowledge transfer by pairing larger teachers with smaller students that differ in depth and width. Across all tasks, SemAlign improves student performance over task matched baselines and over parameter space transfer baselines. Two findings stand out: first, performing latent semantic alignment before any parameter update strongly predicts stable cross-scale transfer; second, steering a small set of paired layers is enough to induce broader behavioral alignment, which makes the method efficient in both compute and data. The replication repository is attached as supplementary material. To summarize, our contributions are as follows:

- We present a semantics-first view of parametric knowledge transfer for cross-scale language models. The formulation treats latent semantic alignment between paired layers as the prerequisite to transfer and uses layer outputs, not raw parameters, as the medium.

- We introduce SEMALIGN, which combines layer attribution and pairing, latent semantic alignment, and representation steering. This design addresses neural incompatibility that limits parameter space transfer in Seeking and LaTen (Zhong et al., 2024; Tan et al., 2025).

- We provide comprehensive experiments on extensive benchmarks with Llama 2. Results and analysis identify the key factors that ease cross-scale transfer and show consistent gains in the efficacy. We provide further discussion in the appendix.

## 2 RELATED WORK

### 2.1 KNOWLEDGE ATTRIBUTION IN LANGUAGE MODELS

Knowledge attribution studies methods for identifying where knowledge resides in large language models and how those components influence predictions. The focus has moved from layer level inspections to neuron level and path level analyses that scale to current models. One representative line designs a static, single pass neuron score that separates "query" and "value" neurons and avoids repeated gradient passes (Yu & Ananiadou, 2024). Moving from units to mechanisms, information flow routes rebuild prediction time computation as a sparse graph and show how influential parts work together during inference (Ferrando & Voita, 2024). In practical analyses, CAPTUM provides operators for layer and neuron attribution, including Internal Influence, Neuron Integrated Gradients, and DeepLIFT or SHAP, which many studies adopt as reproducible baselines (Kokhlikyan et al., 2020). Recent evidence also reports degenerate knowledge neurons, where different neuron sets encode the same fact; this observation supports concept aware or path aware selection when using attribution to guide editing or transfer (Chen et al., 2025).

### 2.2 SEMANTIC ANALYSIS AND LATENT SPACE ALIGNMENT

Semantic analysis and latent space alignment shape and match internal representations so that model adaptation preserves meaning rather than only optimizing an output loss. Within this view, a single research line proposes two connected methods that form a coherent pipeline. Vocabulary Defined Semantics (VDS) uses the model vocabulary to anchor directions in the hidden space and then clusters examples around these anchors, which stabilizes in context learning by better matching data to

the model's internal semantic frame (Gu et al., 2024). Building on that foundation, Semantic Aware Layer Freezing (SALF) treats the structure exposed by VDS as semantic anchors at the layer level and freezes those parts while tuning the remainder, which preserves core semantics and works with parameter efficient finetuning and quantization (Gu et al., 2025). A complementary research thread adjusts hidden states at test time with small edits, showing that behavior can be steered through representation space without heavy retraining (Kong et al., 2024).

### 2.3 PARAMETRIC KNOWLEDGE TRANSFER

Knowledge transfer includes teacher and student distillation, representational matching across layers, and parameter mixing through model merging or task vectors. These approaches provide strong baselines and tools, yet they often work in the output space or assume closely related architectures (Xu et al., 2024; Yang et al., 2024; 2025; Liu et al., 2024). Recent studies frame the problem as parametric knowledge transfer, where the goal is to move internal knowledge that lives inside a model, including parameters and intermediate computations such as activations and residual streams. A representative system, SEEKING, extracts sensitive components from a source, injects them into a target through LoRA initialization, and then applies post alignment fine tuning; results indicate that cross-scale transfer is feasible and that alignment quality is important for stability (Zhong et al., 2024). Follow up work on Neural Incompatibility examines alignment as the main bottleneck cross scales and distinguishes two design choices: PostPKT, which follows extract, inject, and train, and PrePKT, exemplified by LaTen, which aligns parametric spaces with light training before transfer (Tan et al., 2025). Our method adopts semantics-first plan by using latent semantic alignment as a precondition for parametric knowledge transfer, to mitigate neural incompatibility.

## 3 MOTIVATIONAL ANALYSIS

### 3.1 PRELIMINARY: VOCABULARY-DEFINED SEMANTICS

For the recognizable semantic meanings of a given LM, *vocabulary-defined semantics* proposed defining a set of special representations in the latent space to associate with the labels on the vocabulary. It quantifies the semantic property of LM latent space leveraging local isotropy (Cai et al., 2021), and benefits parameter optimizations, such as efficient logits computation (Gu et al., 2024). For each label on the LM vocabulary, there is an associated representation in the latent space, termed as "semantic basis", they share the same semantic meaning, as shown in Figure 1.

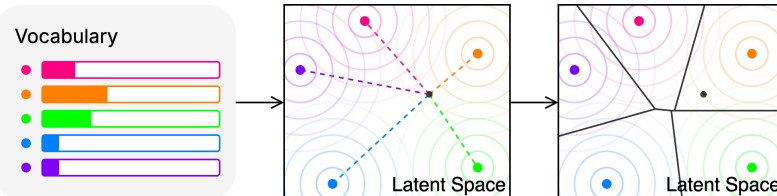

Figure 1: Semantic association of vocabulary and latent space. For each color label on the vocabulary (left), there is a color semantic basis in the latent space (middle). The semantics of the dark dot (indicating an arbitrary representation) in the latent space can be quantified as its cosine similarities to semantic bases. The semantics can be computed as probabilities on the vocabulary. When focusing on the nearest semantic basis for a given latent representation, a latent space can be quantified as discrete semantic regions (right).

For a given LM-head matrix, we conduct matrix multiplication to obtain semantic bases in the latent space. Since the computation direction is from logits to representations, instead of using the LM-head matrix $\mathbb{W}$, we use its pseudoinverse $\mathbb{W}^+$. If there are $v$ labels in the vocabulary, there will be $v$ unique semantic bases representing all semantic meanings. At the output side of LM, we multiply each onehot embedding $\vec{e}$ by the pseudoinverse matrix $\mathbb{W}^+$ to obtain the corresponding representation $\vec{s}$. That is, $\vec{s} = \vec{e} \cdot \mathbb{W}^+$. The computation is equivalent to solving the least squares problem of a system of linear equations. The time cost of computing semantic bases is rather low. For language models like LLaMA 2 (7B, 13B, and even 70B) which has 32000 labels in the vocabulary, it takes around 10 seconds on an A100 GPU. Moreover, this is a one-time computation with persistent value.

## 3.2 EMPIRICAL FINDING: VECTOR NATURE OF SEMANTICS

Centered on each semantic basis, there forms a "semantic field". The concept of semantic field is similar to the *field* term in physics (such as electric field, then the semantic basis analogies to the electric pole). The semantics of an arbitrary latent representation can be quantified as the overlapping impact of numerous semantic fields, and be further computed as probabilities (Gu et al., 2024). The process is "composition of semantics", where multiple *semantic components* become a *resultant vectors* via vector addition. Therefore, we propose a hypothesis that the overlapping effects of semantic fields support a corresponding reversed operation "resolution of semantics". That is, a single *resultant vector* in latent space may be resolved into multiple *component vectors* along the directions of semantic bases.

In detail, for a given latent representation $\vec{r}$, its semantic meaning can be projected to different semantic bases to obtain corresponding semantic components $\vec{c_i} = \mathrm{proj}(\vec{r}, \vec{s_i})$ (analogy to "component force" in a force field). By accumulating the decomposed semantics, we get a "resultant semantics" $\sum_{i=1}^{n} \vec{c_i}$ (analogy to "resultant force" in a force field). The equation $\vec{r} \parallel \sum_{i=1}^{n} \vec{c_i}$ stands approximately true. In contrast, when taking a random collection of vectors as semantic bases and obtain $\vec{c_i'} = \mathrm{proj}(\vec{r}, \vec{s_i'})$, the equation $\vec{r} \perp \sum_{i=1}^{n} \vec{c_i'}$ stays true. It is consistent with the property of the latent space that, arbitrary vectors in a high-dimensional space tend to be orthogonal.

We conduct empirical experiments to validate the hypothesis. For a given data and LM, we first compute the outputs of each layer, and then decompose each layer outputs into semantic components and eventually recompose back as layer outputs. If the old layer outputs amd the new layer outputs share almost similar direction in the latent space, namely their cosine similarity is high, the hypothesis stands. We run with Qwen3 model on HumanEval, to study whether the hypothesis stand with the output-side semantic bases, and using input-side semantic bases for comparison. As shown in Figure 2, the hypothesis stand with the case of using output-side semantic bases because of the very high cosine similarities no matter the layer. In contrast, the situation of using input-side semantic bases is bad and the cosine similarities is close to zero, which indicates the common phenomenon in high-dimensional latent space that arbitrary vectors tend to be orthogonal to each other.

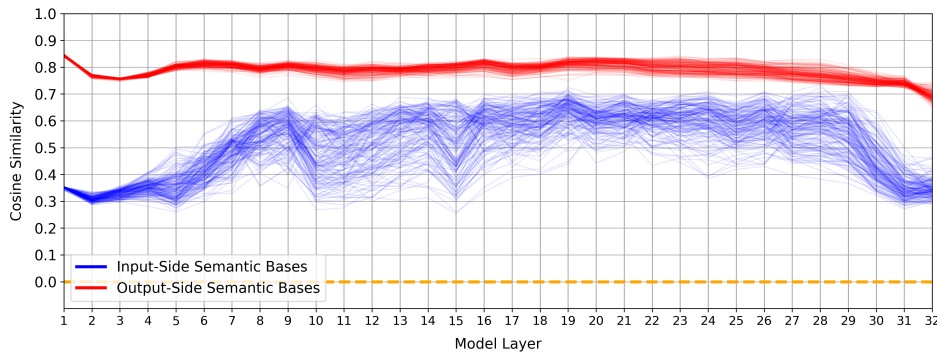

Figure 2: Empirical Validation of Semantics Decomposition on HumanEval with Llama2 (7B).

## 4 APPROACH

Our approach utilized LM semantics to align the latent space between certain layers of teacher and student models. The transferred knowledge by semantic alignment is layer outputs, as the supervisory signal for parameter optimization. We name our approach Semantic Alignment, short as SEMALIGN. The illustration of our approach is in Figure 3, and the main steps are: (1) First, we locate critical layers in teacher LM by attribution algorithm, and find the layer to pair in student LM by pairing strategy. The semantic alignment will happen between the pair layers, from teacher to student; (2) Then, we decompose the semantics of layer outputs in teacher's latent space, and recompose as supervisory layer outputs in student's latent space. It aligns latent spaces while preserving the semantics of layer outputs; (3) Further, in the student model, we optimize layer parameters using

the supervisory layer outputs. For the same given data, the layer outputs of the paired layers become close, indicating the similar behaviors by teacher and student models.

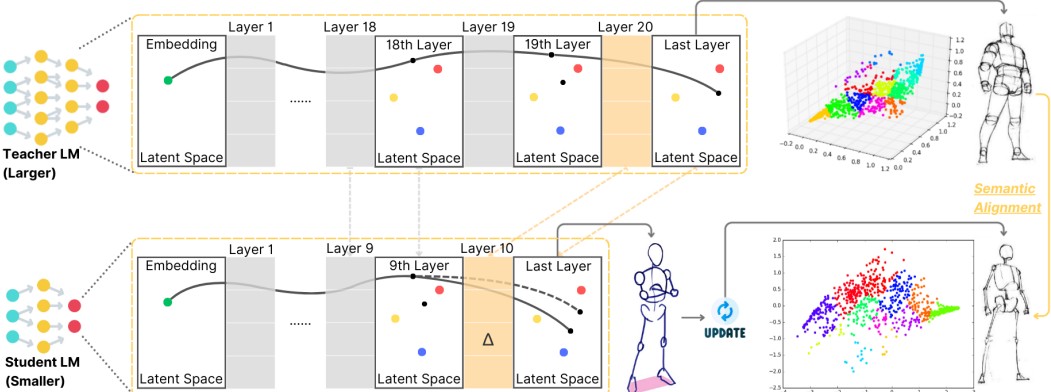

Figure 3: Illustration of our cross-scale knowledge transfer approach. Assume a 20-layer teacher LM and a 10-layer student LM. First, layers in teacher and student models are pairs by dashed arrow lines. Marked by orange color, the 20th teacher's layer is located as critical, and its pair is the 10th student's layer, namely the layer to optimize. Second, represented by the dots in 3D and 2D spaces, the layer outputs from teacher model are decomposed in the larger dimensional teacher's latent space and recomposed in the smaller dimensional student's latent space, as the supervisory signal. It undergos dimensional reduction but still preserves complete semantics, represented by the changes to gray bots, remaining the body gesture but reducing details. Third, the paired student's layer will be updated, to make the student's layer outputs be close to the supervisory signal. It is similar to blue bots, to be adjusted playing the same body gesture as gray bots does. Afte the cross-scale knowledge transfer, student's layer outputs will steer to the supervisory signal, represented by the dashed curve, and partial layer parameters are optimized, marked by the delta symbol. In the last layer's latent space, the model outputs is represented by the small dark dot whose distances to big color dots indicates the probabilities in LM vocabulary. For the teacher LM, its dark dot is close to the red dot, so its output label is the corresponding red label; while for the student LM, its output label was the corresponding blue but will become red after knowledge transfer, since its dark dot moves far away from the blue dot while approaching the red dot.

## 4.1 LAYER ATTRIBUTION AND LAYER PAIRING

**Layer Attribution.** We identify candidate teacher layers using *Layer Gradient × Activation* at the layer level: for each layer, we multiply the gradient of the supervised objective by the layer's activations and aggregate over tokens and channels to obtain a scalar importance score per layer. This choice is simple, stable, and available in standard attribution toolkits (Kokhlikyan et al., 2020).

**Layer Pairing.** Let the teacher have $L_T$ layers and the student have $L_S$ layers. We build a depth-aware mapping that assigns each student layer a teacher counterpart while preserving order. For student index $k \in \{1, \ldots, L_S\}$, define

$$\ell_k^{\mathrm{T}} \;=\; \max\left(1, \left\lfloor \frac{L_T}{L_S}\, k \right\rfloor\right) \in \{1, \ldots, L_T\},$$

which, for example, gives $\ell_k^{\mathrm{T}} = 2k$ when $L_T{=}20$ and $L_S{=}10$. This mapping is one-to-one when $L_T/L_S$ is an integer; otherwise, some teacher layers may be shared across adjacent student layers. If a teacher layer $\ell_\star^{\mathrm{T}}$ is judged critical by attribution but does not coincide with any $\ell_k^{\mathrm{T}}$, we select the nearest *deeper-or-equal* student index

$$k^{\dagger} \;=\; \min\left\{ k' \in \{1, \ldots, L_S\} \;:\; \ell_{k'}^{\mathrm{T}} \geq \ell_\star^{\mathrm{T}} \right\},$$

defaulting to $k^{\dagger}{=}L_S$ if the set is empty. We then operate on the pair $(\ell_{k^{\dagger}}^{\mathrm{T}}, k^{\dagger})$, which preserves depth order and guarantees a concrete student partner for every attributed teacher layer.

If supervision requires a target at an exact student depth while $L_T/L_S$ is non-integer, we interpolate between adjacent teacher layers. Let the student depth be $k^\dagger$ and set $\rho = k^\dagger/L_S$. Define $u = L_T\rho$, then

$$a = \max\big(1, \min(\lfloor u \rfloor, L_T - 1)\big), \quad b = a+1, \quad \lambda = \mathrm{clip}(u-a, 0, 1).$$

Given teacher representations $\mathbf{h}_{(a)}^{\mathrm{T}}$ and $\mathbf{h}_{(b)}^{\mathrm{T}}$ at the supervision interface, the interpolated target for student layer $k^\dagger$ is

$$\tilde{\mathbf{h}}^{\mathrm{T}} = (1-\lambda)\,\mathbf{h}_{(a)}^{\mathrm{T}} + \lambda\,\mathbf{h}_{(b)}^{\mathrm{T}}.$$

This yields a well-defined supervisory signal from the teacher for every student depth while respecting layer ordering.

## 4.2 Latent Semantic Alignment (Teacher → Student)

We construct a training-free, semantics-preserving mapping from a *teacher* layer's latent space to a *student* layer's latent space. The teacher and student come from the same LM family at different scales, and their semantic bases are index-aligned (Section Preliminaries). The idea is to *decompose* the teacher vector into components along teacher semantic atoms and *recompose* those components in the student basis using the *same* coefficients.

Let $\mathbf{h}^{\mathrm{T}} \in \mathbb{R}^{D_T}$ be a teacher-layer output at the supervision interface and $\mathbf{h}^{\mathrm{S}} \in \mathbb{R}^{D_S}$ a student-layer vector (to be constructed as a target). Let $S_{\mathrm{T}} = [\mathbf{s}_1^{\mathrm{T}}, \ldots, \mathbf{s}_m^{\mathrm{T}}] \in \mathbb{R}^{D_T \times m}$ and $S_{\mathrm{S}} = [\mathbf{s}_1^{\mathrm{S}}, \ldots, \mathbf{s}_m^{\mathrm{S}}] \in \mathbb{R}^{D_S \times m}$ denote the teacher and student semantic bases, respectively, with column $i$ in $S_{\mathrm{T}}$ and $S_{\mathrm{S}}$ referring to the same semantic atom. Assume unit-normalized columns: $\|\mathbf{s}_i^{\mathrm{T}}\|_2 = \|\mathbf{s}_i^{\mathrm{S}}\|_2 = 1$.

**Semantics-preserving Decomposition and Recomposition.** We form *semantic coefficients* by cosine projection in the teacher space:

$$\mathbf{a} = \big[\cos(\mathbf{h}^{\mathrm{T}}, \mathbf{s}_1^{\mathrm{T}}), \ldots, \cos(\mathbf{h}^{\mathrm{T}}, \mathbf{s}_m^{\mathrm{T}})\big]^\top = \frac{S_{\mathrm{T}}^\top \mathbf{h}^{\mathrm{T}}}{\|\mathbf{h}^{\mathrm{T}}\|_2} \in \mathbb{R}^m. \tag{1}$$

We then *recompose* in the student space with the same coefficients:

$$\tilde{\mathbf{h}}^{\mathrm{S}} = S_{\mathrm{S}}\,\mathbf{a} \in \mathbb{R}^{D_S}. \tag{2}$$

The vector $\tilde{\mathbf{h}}^{\mathrm{S}}$ serves as the supervisory signal for the student layer output at the same interface.

Because column $i$ of $S_{\mathrm{T}}$ and $S_{\mathrm{S}}$ denote the same semantic atom and all columns are unit-norm, the pair $\big($decompose in teacher: $\mathbf{a} = S_{\mathrm{T}}^\top \mathbf{h}^{\mathrm{T}}/\|\mathbf{h}^{\mathrm{T}}\|_2$, recompose in student: $\tilde{\mathbf{h}}^{\mathrm{S}} = S_{\mathrm{S}}\mathbf{a}\big)$ transfers an identical set of component weights from teacher to student. Thus, the semantic content is preserved; only the ambient basis changes (teacher vs. student).

## 4.3 Cosine-Only Layer and Output Alignment

We adapt the student by updating only the parameters of its paired layer $k$ while freezing all others. The training objective consists of *two* cosine-alignment terms with no additional weights: (i) a *layer-level* cosine loss that aligns the $k$-th layer's activations to the semantics-preserving target, and (ii) an *output-level* cosine loss that aligns the final-layer logits to the supervised label direction. This naming emphasizes that both the intermediate representation (for semantic alignment) and the model output (a geometric surrogate of cross-entropy) are optimized using cosine similarity alone.

Let $\theta^{(k)}$ denote the parameters of student layer $k$. Let $\mathbf{h}^{(k)} \in \mathbb{R}^{B \times T \times D}$ be the student activations at a fixed supervision interface, chosen as the sublayer output *before* residual addition. Let $\mathbf{h}_\star^{(k)} \in \mathbb{R}^{B \times T \times D}$ be the semantics-preserving target instantiated at the same interface (broadcast across batch/tokens as appropriate). Let $\mathbf{z} \in \mathbb{R}^{B \times T \times |\mathcal{Y}|}$ be the final-layer logits, and let $\mathbf{y}^{\mathrm{oh}} \in \{0,1\}^{B \times T \times |\mathcal{Y}|}$ be one-hot (or label-smoothed) target distributions over $\mathcal{Y}$.[1]

The total training objective is an unweighted sum of these two terms: $\mathcal{L}(\theta^{(k)}) = \mathcal{L}_{\mathrm{layer}} + \mathcal{L}_{\mathrm{out}}$. The first term $\mathcal{L}_{\mathrm{layer}} = 1 - \mathrm{Avg}\Big[\cos\big(\mathbf{h}^{(k)}, \mathbf{h}_\star^{(k)}\big)\Big]$ represents the targeted layer's alignment (semantic alignment at layer $k$); while the second term $\mathcal{L}_{\mathrm{out}} = 1 - \mathrm{Avg}\big[\cos\big(\mathbf{z}, \mathbf{y}^{\mathrm{oh}}\big)\big]$ represents the last

---

[1]For instance-level tasks, use per-example logits $\mathbf{z} \in \mathbb{R}^{B \times |\mathcal{Y}|}$ and one-hot targets $\mathbf{y}^{\mathrm{oh}} \in \{0,1\}^{B \times |\mathcal{Y}|}$.

layer's alignment (geometric surrogate of CE at the last layer). Besides, $\mathrm{Avg}[\cdot]$ denotes a simple average over supervised positions, and $\cos(\mathbf{u}, \mathbf{v}) = \dfrac{\langle \mathbf{u}, \mathbf{v} \rangle}{\|\mathbf{u}\|_2 \, \|\mathbf{v}\|_2}$.

The first term, $\mathcal{L}_{\mathrm{layer}}$, enforces *latent semantic alignment*: it drives the $k$-th layer's representation toward the semantics-preserving target constructed by decomposing the teacher's output into teacher semantic components and recomposing them in the student basis. The second term, $\mathcal{L}_{\mathrm{out}}$, aligns the final logits with the ground-truth direction in label space; because cosine similarity is scale-invariant in the logit space, this term acts as a principled geometric surrogate for cross-entropy, encouraging the model to place probability mass on the correct label while avoiding sensitivity to logit scale. Together, the two cosine objectives couple representation-level semantics with output-level correctness using a single, consistent geometric criterion.

We backpropagate through the full network but update only $\theta^{(k)}$. The target $\mathbf{h}_\star^{(k)}$ and the label vectors $\mathbf{y}^{\mathrm{oh}}$ are treated as constants. Matching the supervision interface for $\mathbf{h}^{(k)}$ and $\mathbf{h}_\star^{(k)}$ (both pre-residual) ensures that the alignment term acts directly on the representation controlled by layer $k$, while the output cosine term refines the model's decision geometry at the final layer.

## 5 EXPERIMENTS

In the experiments, we mainly study how our approach performs in parameteric knowledge transfer comparing with PKT baselines. We also conduct analysis based on the latent representation similarities between teacher and student's models before and after latent semantic alignment.

### 5.1 SETUP

**Datasets.** We use four well-established benchmarks, covering the most common downstream tasks: MMLU measures professional knowledge (Hendrycks et al., 2021); GSM8K measures mathematical reasoning (Cobbe et al., 2021); HumanEval and MBPP measure code generation (Chen et al., 2021; Austin et al., 2021).

**Models.** We conduct experiments with Llama 2 (Touvron et al., 2023) models, mainly chat versions instead of base versions for the better instruction-following ability. Besides, we employ LM variants to study the transfer from further-finetuned teacher models to a same student model, CodeLlama-13B-Python (Roziere et al., 2023) and WizardCoder-13B-Python (Luo et al., 2023). They are fine-tuned on Llama-2-13B with massive code data for an enhanced coding performance.

**Metrics.** For MMLU and GSM8K, we calculate *accuracy* in zero-shot setting; and for HumanEval and MBPP, we calculate *pass@1*. Larger scores mean better performance.

**Baselines.** The prior work on parametric knowledge transfer is SEEKING and LATEN, which are the baselines in our experiments. Both perform parametric knowledge transfer in two stages: Seeking is PostPKT (inject-then-train) while LaTen is PrePKT (align-then-inject). SEEKING first *extracts* task-relevant parameters from a teacher by ranking weights via sensitivity scores (gradient×parameter on a seed set), then *injects* them into a student. Layer-wise importance is aggregated to pick the top layers; within each selected matrix, a high-sensitivity sub-block is chosen to bridge width/depth gaps. Each extracted block is SVD-factorized to initialize a low-rank LoRA $(B, A)$, after which the student is post-aligned by standard fine-tuning. LATEN adopts a *Locate-Then-Align* pipeline to minimize post-training. It *locates* neuron-level carriers of knowledge in FFN/MHSA using static attribution, selects top neurons per layer, and forms teacher-side deltas. A lightweight hypernetwork $g_\phi$ is trained on a tiny alignment set to *pre-align* these deltas into the student's parameter shape/scale, which are then injected once for immediate gains. This design targets cross-model incompatibilities by aligning deltas before injection rather than SVD-to-LoRA initialization plus post-alignment.

### 5.2 RESULTS

**Results of Cross-Scale Knowledge Transfer.** In all four benchmarks, SEMALIGN improves substantially over Llama2-7B-Chat while remaining below the Llama2-13B-Chat teacher, and it stays closer to the teacher than the other transfer baselines on average. Concretely, the absolute gaps

between SemAlign and 13B are 2.60 (MMLU: 50.30 vs 52.90), 1.34 (GSM8K: 19.21 vs 20.55), 1.41 (HumanEval: 17.34 vs 18.75), and 0.42 (MBPP: 18.78 vs 19.20), averaging 1.44 points. This average gap is smaller than SEEKING ($\approx$ 3.92) and LATEN ($\approx$ 3.43). A per-task view shows one exception—on GSM8K, LaTen (20.47) is numerically closest to 13B (20.55)—while SEEKING overshoots the teacher (28.23). Overall, SemAlign's three-of-four closer margins indicate it learns the teacher's behavior more faithfully than the baselines.

| Models | MMLU | GSM8K | HumanEval | MBPP |
|---|---|---|---|---|
| Llama2-7B-Chat | 44.20 | 16.07 | 14.05 | 17.80 |
| Llama2-13B-Chat | 52.90 | 20.55 | 18.75 | 19.20 |
| Seeking | 49.60 | **28.23** | 15.44 | **20.60** |
| LaTen | 44.40 | 20.47 | 14.63 | 18.20 |
| **SemAlign** | **50.30** | 19.21 | **17.34** | 18.78 |

Table 1: Results of Parametric Knowledge Transfer in Diverse Downstream Tasks.

On task leadership, SemAlign attains the best transferred performance on MMLU (50.30) and HumanEval (17.34), surpassing both Seeking (49.60, 15.44) and LATEN (44.40, 14.63), whereas SEEKING leads on GSM8K (28.23) and MBPP (20.60). Notably, SEEKING exceeds the 13B teacher on both GSM8K (+7.68) and MBPP (+1.40), while SemAlign remains below but close to the teacher; this pattern is consistent with SemAlign's cosine-similarity objective encouraging conservative matching of teacher representations, whereas SEEKING appears to incorporate additional parameter optimization beyond pure transfer.

An additional observation is the stability–aggressiveness trade-off across methods: SEEKING achieves large gains on reasoning- and coding-flavored datasets by overshooting the teacher (GSM8K, MBPP), suggesting stronger task-specific optimization, while SemAlign stays within $\leq 2.60$ points of the teacher on every task, indicating steadier transfer that narrows the gap without over-amplifying particular skills.

**Results of Knowledge Transfer from Finetuned Models.** SEMALIGN consistently outperforms the transfer baselines in five of six teacher–task settings, indicating stronger parametric knowledge transfer. With Llama2-13B-Chat as teacher, it leads LATEN and SEEKING on HumanEval (17.34 vs 14.63/15.44), and only trails SEEKING on MBPP (18.78 vs 20.60). The advantage becomes clearer with code-specialized teachers: from CodeLlama-13B-Python, SemAlign reaches 20.12 (+4.07 over SEEKING, +6.10 over LATEN) on HumanEval and 22.35 (+0.95, +4.55) on MBPP; from WizardCoder-13B-Python, it attains 19.46 (+4.42, +5.44) and 21.18 (+1.38, +2.58) on HumanEval and MBPP, respectively. These trends show that SemAlign extracts and transfers teacher competency more reliably, especially when the teacher is stronger in coding.

| Models | HumanEval | MBPP |
|---|---|---|
| Llama2-7B-Chat | 14.05 | 17.80 |
| Llama2-13B-Chat | 18.75 | 19.20 |
| Seeking | 15.44 | **20.60** |
| LaTen | 14.63 | 18.20 |
| **SemAlign** | **17.34** | 18.78 |
| CodeLlama-13B-Python | 47.56 | 37.80 |
| Seeking | 16.05 | 21.40 |
| LaTen | 14.02 | 17.80 |
| **SemAlign** | **20.12** | **22.35** |
| WizardCoder-13B-Python | 56.71 | 41.60 |
| Seeking | 15.04 | 19.80 |
| LaTen | 14.02 | 18.60 |
| **SemAlign** | **19.46** | **21.18** |

Table 2: Results of Parametric Knowledge Transfer from Finetuned Teacher Models.

Across two coding benchmarks, all methods remain far below the finetuned teachers (CodeLlama-13B-Python: 47.56/37.80; WizardCoder-13B-Python: 56.71/41.60), despite often surpassing Llama2-7B-Chat and sometimes even matching or exceeding Llama2-13B-Chat (e.g., SEEKING on MBPP with 20.60 vs 19.20). This gap suggests that the extensive, task-specific optimization baked into code-specialized teachers is difficult to reconstruct via short-horizon transfer; objectives like cosine matching encourage conservative alignment to teacher representations rather than aggressive task re-optimization, limiting the attainable ceiling without longer or more targeted finetuning.

An additional observation is that SEEKING only overshoots the teacher on MBPP when the teacher is the generalist Llama2-13B-Chat (20.60 > 19.20) but not when the teacher is code-specialized; meanwhile, SEMALIGN shows its largest margins over baselines precisely when transferring from code-specialized teachers. This pattern hints that aggressive, task-specific optimization in SEEKING can exploit headroom left by generalist teachers, whereas SemAlign's representation-faithful transfer scales better with teacher specialization.

## 5.3 ANALYSIS

We adopt Centered Kernel Alignment (CKA) (Kornblith et al., 2019) as the analysis tool to study the similarities between layer outputs from teacher and student models. We run Llama2-chat models on HumanEval data. CKA is commonly used to compute the similarities between feature representations in neural networks, which is based on Hilbert-Schmidt Independence Criterion (HSIC).

As shown in Figure 4, there are high similarities between the layer outputs from teacher and student models, especially along the main diagonal. It indicates that, there exists no neuron incompatibility if using layer outputs as the medium of parameteric knowledge transfer, instead of directly using layer parameters. The highest similarities is almostly layer-by-layer, from shallow to deep. Meanwhile, the cases before (the left subfigure) and after (the right subfigure) latent semantic alignment share very similar pattern of similarities. It means, adopting latent space alignment is a safe way to utilize the similarities between layer outputs between cross-scale language models.

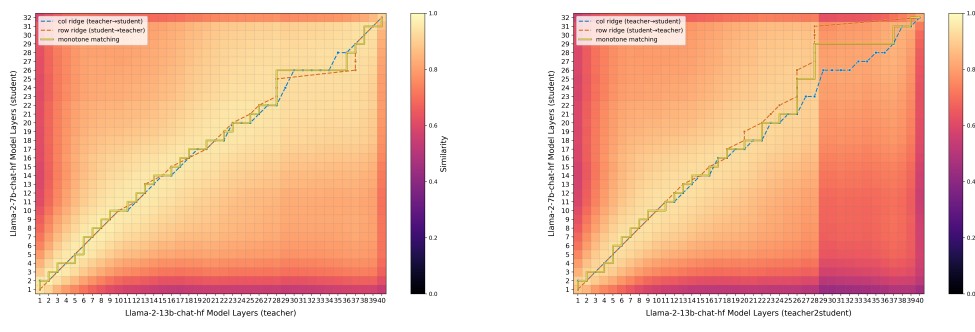

(a) teacher-student w/o alignment       (b) teacher-student w/ alignment

Figure 4: Comparison of Layer-wise Representation Similarities between LLMs.

## 6 CONCLUSION

We studied parametric knowledge transfer across differently scaled LLMs from a *semantics-first* perspective. Rather than moving raw parameters as in prior paradigms, we use layer outputs as the medium of transfer and identify *latent semantic alignment* as the prerequisite for stable cross-scale transfer. Building on this view, SEMALIGN locates and pairs teacher and student layers, aligns their latent semantics, and then steers the student so its paired layers reproduce the aligned supervisory hidden states. This design avoids neural incompatibility, simplifies the procedure, and makes transfer efficient in both compute and data. Empirically, SemAlign improves students over task-matched baselines and over parameter-space transfer methods on four benchmarks with Llama 2 families. The results support our central claim: treating activations as the carrier of knowledge and aligning semantics-first provides a robust path to cross-scale PKT.

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

## A    IMPLEMENTATION DETAILS

### A.1    STATS OF LANGUAGE MODELS

The stats of language models in our experiments are shown in Table 3.

|  | Llama2 | | CodeLlama | WizardCoder |
|---|---|---|---|---|
|  | 7B | 13B | 13B | 13B |
| Head Num. | 32 | 40 | 40 | 40 |
| Layer Num. | 32 | 40 | 40 | 40 |
| Dimension | 4,096 | 5,120 | 5,120 | 5,120 |
| Vocabulary | 32,000 | 32,000 | 32,000 | 32,001 |

Table 3: Stats of Llama 2 Language Models.

## A.2 IMPLEMENTATIONS DETAILS

We follow the experimental protocol of LATEN for a fair comparison, but unlike LaTen, our approach uses a single training phase only: the alignment operation is integrated into the training objective as an auxiliary loss term, with no separate alignment stage or post-training optimization. In detail, we fine-tune the smaller model for 5 epochs with a batch size of 64 and a learning rate of $3 \times 10^{-4}$ (for HUMANEVAL, $3 \times 10^{-5}$), and use 3 epochs in the SFT setting; LoRA uses rank $r{=}16$ and is inserted into FFN (`up_proj`, `down_proj`) and MHSA (`v_proj`, `o_proj`) modules. For baseline reproduction under LATEN's protocol, the hypernetwork is trained with a learning rate of $1 \times 10^{-5}$ and weight decay 0.05, the sample size is $P{=}16$, and 10% of neurons are transferred per layer. The hyperparameters on alignment and trainment are shown in Table 4 and Table 5.

Our implementation uses deep learning framework PYTORCH (Paszke et al., 2019), TRANSFORMERS (Wolf et al., 2019) and vLLM (Kwon et al., 2023).

|  | MMLU | GSM8K | HumanEval | MBPP |
|---|---|---|---|---|
| Steps | 2 | 4 | 3 | 8 |
| Align Size | 32 | 64 | 48 | 128 |
| Learning Rate | 3e-5 | 3e-5 | 3e-5 | 3e-5 |

Table 4: Implementation Details in Alignment.

|  | MMLU | GSM8K | HumanEval | MBPP |
|---|---|---|---|---|
| Epochs | 5 | 5 | 3 | 5 |
| Train Size | 1000 | 1000 | 1000 | 300 |
| Learning Rate | 3e-4 | 3e-4 | 3e-5 | 3e-4 |

Table 5: Implementation Details in Training.

## B DETAILS OF PARAMETRIC KNOWLEDGE TRANSFER BASELINES

Both baselines view knowledge as model weights and use the same two-step process: *extract* from the teacher, then *inject* into the student. They also handle layer/width mismatches and are tested on multiple LLM benchmarks. SEEKING focuses on sensitivity-based selection and LoRA initialization, followed by post-training alignment, so its alignment cost comes after injection but yields stable gains. LATEN focuses on neuron-level localization and hypernetwork pre-alignment, shifting the cost upfront to reduce or avoid post-training; in doing so, it highlights neural incompatibility and motivates semantics-first alignment. The detailed technical descriptions are as follows:

### B.1 ILLUSTRATE SEEKING

SEEKING treats parametric knowledge transfer as two stages: *extract* task-related parameters from a larger teacher, then *inject* them into a smaller student and perform *post-alignment* fine-tuning. Given a task $\mathcal{T}$ and a small *seed* set produced by the teacher (typically a few dozen decoded examples),

SEEKING assigns an importance score to each teacher parameter $\theta_i$ via *sensitivity*:

$$S_{i,j}^{\mathcal{T}} = \left| \theta_i^\top \nabla_{\theta_i} \mathcal{L}(x_j^{\mathcal{T}}, y_j^{\mathcal{T}} \mid \Theta) \right|, \qquad S_i^{\mathcal{T}} = \sum_{j=1}^{k} S_{i,j}^{\mathcal{T}},$$

a first-order approximation of the loss increase if $\theta_i$ were removed. Layer scores are obtained by summing parameter sensitivities within the layer; the top $L_s$ layers (order-preserving) are kept for transfer. To bridge depth/width mismatches, SEEKING performs *sensitivity-guided dimensionality reduction* on each selected weight matrix $W^l \in \mathbb{R}^{n_l \times m_l}$ by choosing a submatrix $W_{\text{extract}}^l \in \mathbb{R}^{n_s \times m_s}$ (rows/columns or 2D block) that maximizes cumulative sensitivity:

$$W_{\text{extract}}^l = \arg \max_{W' \subseteq W^l} \sum_{\theta_i \in W'} S_i \quad \text{s.t.} \quad n_s \leq n_l, \ m_s \leq m_l.$$

The extracted blocks across layers are aggregated into $\Delta\Theta_{\text{extract}}$. For *injection*, each $W_{\text{extract}}^l$ is factorized with SVD, $U\Sigma V^\top$, to initialize a rank-$r$ LoRA pair $(B, A)$ via $B \leftarrow U_{[:,1:r]} \Sigma_{1:r,1:r}$ and $A \leftarrow V_{1:r,:}^\top$, yielding an initialized student

$$W^{l\star} = W^l - W_{\text{extract}}^l + BA,$$

after which the student is fine-tuned to align the injected deltas. Empirically, SEEKING reports consistent gains across reasoning, professional knowledge, instruction-following, and open-domain dialogue, and analyzes factors such as teacher scale, initialization strategy, seed count, module origin, and LoRA rank.

## B.2 ILLUSTRATE LATEN

LATEN formalizes two PKT regimes: *PostPKT* (inject then train-to-align) and *PrePKT* (align then inject). It proposes *Locate-Then-Align* to reduce or avoid post-training. For a larger teacher $M_\ell$ with parameters $\Theta_\ell$ and a smaller student $M_s$ with $\Theta_s$, LaTen first *locates* the most informative sites at *neuron* granularity, then learns a light *hypernetwork* to *pre-align* teacher deltas to the student's parameter space before injection:

$$\Delta\Theta_\ell \leftarrow \text{Locate}(M_\ell; \mathcal{D}_{\text{extract}}), \qquad \widehat{\Delta\Theta}_s \leftarrow g_\phi(\Delta\Theta_\ell), \qquad \Theta_s^\star \leftarrow \text{Inject}(\Theta_s, \widehat{\Delta\Theta}_s).$$

Using a static neuron-level attribution method, LaTen scores neurons in both FFN and MHSA (per selected layer; last-useful-token based) and selects the top-$k$ neurons per layer for transfer, motivated by evidence that neurons serve as units storing skills/knowledge; this yields vectorized teacher deltas $\Delta\Theta_\ell$ over chosen FFN/MHSA submodules.

To bridge width/depth gaps and value-scale discrepancies, a small two-layer MLP *hypernetwork* $g_\phi$ (with ReLU) is trained on a tiny *alignment* set (often $< 100$ examples) to map teacher deltas into student-shaped deltas by minimizing the LM loss while the base weights remain frozen:

$$\min_\phi \ \mathbb{E}_{(x,y) \in \mathcal{D}_{\text{align}}} \mathcal{L}_{\text{LM}}\big(y; \ M_s(x; \ \Theta_s \oplus g_\phi(\Delta\Theta_\ell))\big).$$

After learning $g_\phi$, $\widehat{\Delta\Theta}_s$ is injected once, aiming for immediate gains without further training. LaTen contrasts this with SEEKING's SVD-to-LoRA initialization, and attributes transfer instability to *neural incompatibility* (low similarity across behavioral and parametric spaces) when deltas are unaligned. Experiments show promising (though not uniformly stable) improvements under PrePKT and comparisons against baselines such as self-derived LoRA and language-based distillation.

## C DISCUSSION

### C.1 MEDIUM IN PARAMETRIC KNOWLEDGE TRANSFER.

Compare with the prior work such as Seeking and LaTen, which take model weights as the medium for knowledge transfer, SEMALIGN suggest using layer outputs as the medium. Our approach shows

advantages in efficacy, and also, requires almost no computation cost for alignment. Moreover, our methodology have theorically better performance based on the following reasons.

For the perspective of information transfer, layer outputs as the medium requires less bandwidth than the prior work of PKT, as well as the general knowledge transfer work. Because in language models, the dimension size of layer outputs is much smaller than that of layer parameters, as well as that of the probabilities in LM vocabulary. A smaller size indicates less information to transfer. Therefore, transferring more knowledge of same quality requires more computation cost; or, transferring more knowledge by costing same computation indicates high information loss.

From the perspective of the association between knowledge and neurons, layer outputs is a better choice than layer parameters. It is known by prior work that LM knowledge and neurons follow many-to-many dynamic associations (Allen-Zhu & Li). Therefore, if knowledge transfer is conducted through layer parameters (especially LaTen), certain student's parameters will be updated with certain teacher's parameters. However, no matter the layer parameters from the teacher or student model, they associate with not only the current given data, but also other data. Such practice of parametric knowledge transfer by direct parameter manipulation is likely to cause potential side effects. In contrast, Seeking indicates a safer practice, which introduces the idea of parametric knowledge transfer to the framework of parameter-efficient finetuning.

## C.2 LIMITATIONS AND FUTURE WORK

Our study focuses on a limited set of tasks, and layer-level pairing, while broader coverage (architectures, modalities, safety-critical settings) remains open. Future directions include: (1) extending semantic alignment to finer granularity (sub-layer, attention head) (2) comparing objectives that combine causal and semantic constraints; (3) exploring better strategies on layer pairing based on the layer outputs in teacher and student models; (4) scaling analyses across families with larger architectural gaps to stress-test robustness. We hope SemAlign serves as a simple, practical foundation for activation-driven knowledge transfer and as a stepping stone toward precise, low-loss "brainwave" communication between models.

