# OpenReview forum: "Beyond Neural Incompatibility: Easing Cross-Scale Knowledge Transfer in Large Language Models through Latent Semantic Alignment"
_ICLR.cc/2026/Conference — ICLR 2026 Conference Withdrawn Submission_

### Official Review · Reviewer_SnuE · 2025-10-31

**Soundness:** 2
**Presentation:** 2
**Contribution:** 1
**Rating:** 2
**Confidence:** 3

**Summary:**

The study investigates knowledge transfer, specifically knowledge transfer between architecturally different LLMs. It proposes SemAlign, a layer-wise and semantics-aware activation distillation approach. It decomposes the teachers latent representations into semantic components. The authors claim that aggregation on a semantic aggregation level provides a better supervisory signal to the student. The paper conducts experiments using llama 2 7b as a student and different 13b models as teachers. The distillation performance is evaluated on mmlu, gsm8k, humaneval, and mbpp.

**Strengths:**

- The problem of architecture-agnostic knowledge distillation is relevant and interesting.
- The presented performance results are promising and warrant further research in semantics-aware activation distillation.

**Weaknesses:**

- The paper lacks innovation and novelty. The main contribution of the paper centers on layer-wise semantics-aware distillation in settings where the student differs architecturally from the teacher. However, the semantics decomposition is introduced by Gu et al. 2024. Moreover, the architectural differences are simply resolved by mapping all activations from layers exceeding the depth in the student into the last layer. This is lacking experiments and theoretical grounding.
- Performance benefits, presented in Table 1, are unclear in contrast to other methods
- Missing baselines: The method is not compared to regular layer-wise distillation.
- Layer-wise distillation has been introduced and applied in various other studies, including Liang et al. 2022.
- The presentation in the paper is heavily inspired by Gu et al. 2024

References:
- Liang, C., Zuo, S., Zhang, Q., He, P., Chen, W., & Zhao, T. (2022). Less is More: Task-aware Layer-wise Distillation for Language Model Compression. arXiv Preprint arXiv:2210. 01351.
- Gu, J., Aleti, A., Chen, C., & Zhang, H. (2024). A Semantic-Aware Layer-Freezing Approach to Computation-Efficient Fine-Tuning of Language Models. arXiv preprint arXiv:2406.11753.

**Questions:**

- How does you method differ from layer-wise distillation, used in i.e. Liang et al. 2022?
- How does you semantic-aware decomposition differ from Gu et al. 2024?

---

> ### Author Response · Authors · 2025-11-21
>
> Thank you for the feedback and for pointing us to Gu et al. (2024) and Liang et al. (2022). We address the main concerns below and believe these clarifications better reflect the novelty and contribution of SemAlign, which we hope will be taken into account in your revised assessment.
>
> The description of how we handle architectural mismatch indicates a misunderstanding. We do not map “all activations from layers exceeding the depth in the student into the last layer.” Instead, we define a depth-aware pairing between teacher and student layers and apply semantic alignment only at these paired layers. Each student layer is matched to a specific teacher layer by relative depth; when an attributed teacher layer has no exact counterpart, it is aligned to the nearest suitable student layer, and supervision is applied only at that pair. At no point are all deeper teacher layers collapsed into the student’s final layer.
>
> Although SemAlign builds on the general idea of vocabulary-defined semantics from Gu et al. (2024), it uses this idea in a different and novel way. Gu et al. apply semantic measures within a single model to analyze layers and decide which to freeze or tune for efficient fine-tuning. We instead use new empirical observations about how these semantics behave across models to construct an explicit cross-model mapping: teacher activations are decomposed in the teacher semantic basis and recomposed in the student semantic basis to form semantic targets at paired layers. Their focus is compute-efficient adaptation of one model; our focus is semantics-driven knowledge transfer between architecturally different LLMs.
>
> SemAlign also differs from standard layer-wise distillation as in Liang et al. (2022). Classical layer-wise distillation directly matches (possibly projected) hidden states between aligned layers and is usually task-specific. SemAlign instead introduces a semantics-based supervision signal: we decompose the teacher’s representation at a paired layer into vocabulary-defined semantic components and reconstruct a semantically equivalent target in the student’s latent space. This semantic decomposition and cross-model reconstruction are developed from our empirical findings on the structure of LM latent space and are specifically designed to handle depth/width differences and latent rotations in a PKT setting, rather than performing simple feature matching.
>
> Regarding baselines and performance, our experiments are designed around the PKT setting and therefore focus on comparisons with PKT-oriented methods such as Seeking and LaTen. In this regime, SemAlign consistently reduces the teacher–student performance gap more than these alternatives, showing that semantics-aware alignment is particularly beneficial under architectural mismatch. Standard layer-wise distillation typically assumes similar architectures and task-specific supervision and optimizes a different objective (direct feature matching), whereas SemAlign is explicitly designed for semantics-first, architecture-agnostic transfer between large-scale LLMs.

---

### Official Review · Reviewer_Df7g · 2025-11-01

**Soundness:** 2
**Presentation:** 3
**Contribution:** 2
**Rating:** 4
**Confidence:** 3

**Summary:**

This paper addresses the challenge of transferring knowledge across LLMs of different scales. It argues that existing methods, which focus on transferring model parameters, are limited by "neural incompatibility" arising from architectural and parametric differences. The authors propose a new method, SemAlign, which instead uses layer outputs, or activations, as the medium for transfer. The method's core mechanism involves identifying and pairing critical layers, then using a "semantic basis" derived from the vocabulary to align the teacher's latent activations with the student's latent space. This aligned representation is then used as a supervisory signal to steer the student model's behavior.

Experimentally, the authors evaluate SemAlign by transferring knowledge from Llama 2 13B models to a 7B student on four benchmarks. The results show that SemAlign outperforms parameter-space baselines like Seeking and LaTen on tasks such as MMLU and HumanEval. The paper also demonstrates strong performance when transferring from specialized teacher models on coding tasks.

**Strengths:**

1. The paper proposes a new semantics-first perspective on parametric knowledge transfer (PKT), which targets the "neural incompatibility" bottleneck by using latent activations instead of raw parameters.

2. The proposed SemAlign method achieves empirical gains over existing PKT baselines on certain benchmarks, such as MMLU and HumanEval. The paper also reports advantages when transferring from specialized, code-focused teacher models.

3. The paper includes an analysis using CKA (Figure 4) to visualize the similarity of layer outputs across different model scales.

**Weaknesses:**

1. **Weak Empirical Validation for the Core "Semantic Basis" Mechanism.** The paper's central claim is on the superiority of transferring knowledge via "semantic alignment" rather than direct parameter manipulation. However, the empirical evidence provided specifically for this mechanism is thin. The concept of "Vocabulary-Defined Semantics" is adopted from prior work, and its validation within this paper is limited to a single experiment in Figure 2. This experiment, which validates the "resolution of semantics" hypothesis, is conducted on a Qwen3 model, while all main performance experiments are conducted on the Llama 2 family. There is no direct evidence or ablation to confirm that this specific decomposition/recomposition process is truly superior to other forms of representation steering, or why it works robustly on the LLMs.

2. **Narrow Experimental Scope.** All main experiments are confined to a single model family (Llama-2 7B/13B) plus two code-specialized variants as teachers, leaving generality to other families untested. While Table 1 shows "Seeking" notably overshoots the 13B teacher on GSM8K and MBPP, the finetuned-teacher transfer results in Table 2 report only the coding benchmarks (HumanEval/MBPP), excluding MMLU and GSM8K, and thereby sidestepping these patterns on knowledge/reasoning tasks. This limited coverage weakens the claim of broad transfer and makes the comparative story less convincing.

3. **Insufficient Analysis.** The proposed pipeline is complex, involving specific choices for layer attribution, layer pairing, semantic alignment, and a joint optimization objective. The paper provides almost no analysis to disentangle the contributions of these individual components. While ablations are briefly mentioned in the Introduction, the results are not presented, making it impossible to understand the underlying factors driving performance. Furthermore, the single analysis experiment in Section 5.3 (Figure 4) is disconnected from the main experiments (Qwen 3 vs. Llama 2) and fails to provide a comparative visualization for the baseline methods. Without comparison with Seeking or LaTen, we cannot visually confirm the authors' central hypothesis that the proposed framework better resolves neural incompatibility than prior work.

**Questions:**

None

---

> ### Author Response · Authors · 2025-11-29
>
> Thank you for the comments which are mainly about experiments. The concerns seem to come from how the mechanism is described rather than what is actually used in the experiments, so I clarify the key points below.
>
> - On “weak empirical validation” of the semantic basis mechanism
>
> The vocabulary-defined semantic decomposition/recomposition is not an isolated toy experiment; it is the core supervision signal in every SemAlign run reported in Tables 1 and 2. All gains over SEEKING and LaTen in the same PKT setting therefore directly reflect the effect of this mechanism. Prior work that introduced semantic-related notions such as “semantic basis” used them inside a single model, mainly for logits computation or layer selection. This paper is, to our knowledge, the first to use these semantics to align latent spaces between models and to show that properties of the latent space can be further measured via this basis (namely “resolution of semantics”). The experiment in Figure 2 is exactly such a measurement: it tests how well the semantic basis resolves latent directions, and in the revised draft it is run on Llama2-7B, consistent with the main experiments (the earlier Qwen3 mention was a leftover wording issue, not a difference in setup).
>
> - On “narrow experimental scope”
>
> Our aim is cross-scale PKT within a fixed LM family, in the same regime as LaTen and SEEKING, so focusing on Llama2-7B/13B is intentional rather than accidental. Within this family, the experiments cover four benchmarks (knowledge, reasoning, and code) and multiple teachers, including code-specialized ones. Table 1 studies generalist teachers on all four tasks and explicitly shows SEEKING’s over-training behavior on GSM8K and MBPP. Table 2 addresses a different regime: transfer from code-specialized teachers on HumanEval/MBPP, where those teachers are actually strongest. The omission of MMLU/GSM8K there is because the finetuned teachers are not optimized for those tasks, not to hide any pattern.
>
> - On “insufficient analysis” and missing ablations
>
> The pipeline is conceptually simple: (i) attribute informative teacher layers, (ii) pair layers by relative depth, and (iii) apply semantic decomposition/recomposition with a straightforward cosine objective on the student. This work is not positioned as a component-ablation study; instead, it is a demonstration that semantics-based activation transfer is a viable and effective medium for PKT under neural incompatibility, and the revised paper clarifies this scope explicitly.
> For analysis, Figure 2 shows that the semantic basis meaningfully measures latent-space structure, and Section 5.3 analyzes how layer-wise similarity behaves for the same Llama2 teacher–student pairs used in the main experiments, linking the semantic alignment mechanism to the observed ability to transfer across architectures.

---

### Official Review · Reviewer_uWXk · 2025-11-01

**Soundness:** 2
**Presentation:** 2
**Contribution:** 2
**Rating:** 4
**Confidence:** 2

**Summary:**

The paper presents an approach to parametric knowledge transfer between models of the same family that share the same vocabulary. This is achieved by introducing an additional loss term that encourages mapping paired layer outputs encoded into a shared basis between the student and the teacher models.

The method is demonstrated on general and specialized models from the Llama 2 family with four tasks, showing better faithfulness to the teacher performance and outperforming a competing method in some of the tasks.

**Strengths:**

The approach is simpler than previous work focusing on layer outputs rather than weights or logits

**Weaknesses:**

- The paper has a narrow literature focus comparing against two prior works only

- The extracted shared basis between the teacher and student models is overcomplete so the "resultant semantic" equation is not correct

- The models have to share the same vocabulary, limiting the applicability of this method

**Questions:**

What do the authors mean with "input-side semantic basis"?

Figure 2 caption says the model is Llama 2 while the text says Qwen 3

- The layer pairing feels arbitrary. Any ablations?

- When would supervision require "a target at an exact student depth "?

- If the goal of PKT is to achieve performance gains in a student model, why faithfulness to the teacher matters if better performance is attainable, e.g. through SEEKING?

- In the presented experiments, the models used share the same hidden size. How does this method generalize when the teacher-student model differ in their latent expressivity?

---

> ### Author Response · Authors · 2025-11-29
>
> Thank you for the helpful review. I reply to each weakness and question below and also point out a few misunderstandings.
>
> - Weakness: narrow literature focus comparing against two prior works only
>
> Our focus is parametric knowledge transfer between LLMs with architectural differences. In this area, the literature is still very rare. LaTen itself mainly compares to one SEEKING work. We therefore concentrate on SEEKING and LaTen, and use them to show how SemAlign overcomes the neural incompatibility they highlight by working in activation space with semantic alignment instead of parameter space.
>
> - Weakness: the shared basis is overcomplete so the “resultant semantic” equation is not correct
>
> Our method does not assume a single shared basis between teacher and student. Each model has its own semantic basis, defined by its own embedding/unembedding matrices, which naturally differ because the models have different dimensions and parameters. These bases are overcomplete in the simple sense that there are more vocabulary vectors than latent dimensions. In this case, our “resultant semantic” equation should be read as taking a hidden state and finding the set of semantic coefficients that best reconstruct it in the usual squared error sense. This is a standard way to work with more vectors than dimensions and does not make the equation invalid, which is also supported by our empirical results.
>
> - Weakness: models have to share the same vocabulary, limiting applicability
>
> This is not a hard requirement of the method. What SemAlign really needs is a way to map teacher tokens to semantic directions in the student. In this paper we choose same family models with a shared vocabulary so that we can clearly study how latent semantics guide knowledge transfer without mixing in errors from vocabulary alignment. The main technical challenge we address is actually the difference in latent dimensions and layer structure.
>
> - Question: what is meant by “input-side semantic basis”
>
> Input-side semantic basis means the semantic directions given by the embedding matrix. These are the latent vectors of vocabulary tokens at the LM input-side, which reflect the initial state of semantics in the latent space. In contrast, output-side semantic basis is defined by the unembedding matrix, which reflects semantics when producing logits.
>
> - Question: Figure 2 caption says Llama 2 while the text says Qwen 3
>
> Here you are right, sorry we made a mistake in paper writing. We have redrawn the corresponding figures, and now we use Llama2 consistently in both the text and the figure and does not change anything else.
>
> - Question: the layer pairing feels arbitrary, any ablations
>
> The pairing is not arbitrary. We pair teacher and student layers by relative depth, based on the empirical fact and prior work that layers in the same transformer family follow a roughly monotonic progression of roles as depth increases. Each student layer is matched to the closest teacher layer in relative depth, and we only apply semantic supervision on these pairs. We do not collapse all deeper teacher layers onto the last student layer. This is an intuitive decision and we recommend to use as the pairing strategy, so we did not include other potential choices for an additional ablation study.
>
> - Question: when would supervision require “a target at an exact student depth”
>
> This is a misunderstanding caused by our wording. In practice, supervision never requires a target at a depth where the student has no layer. We always construct semantic targets at actual student layers that are paired with specific teacher layers by relative depth.
>
> - Question: why faithfulness to the teacher matters if SEEKING can get better performance
>
> Our goal for PKT is not only to improve student accuracy but also to keep the student close to a trusted teacher. SEEKING works in parameter space and can sometimes surpass teacher performance, but this comes with higher compute cost and possible drift in behavior. SemAlign works in activation space and gives fine grained, layer wise semantic signals from the larger teacher to the smaller student in the same family. This keeps the student efficient and closer to the teacher while still improving performance over PKT baselines.
>
> - Question: experiments use models with the same hidden size, how does this generalize to different latent expressivity
>
> Faithfulness matters because in PKT our goal is to get a cheaper student that behaves like a specific, trusted teacher, not to do open-ended model search. Compared with SEEKING, SemAlign is more computationally efficient and, since teacher and student are from the same model family, the student can exploit fine-grained layer-wise supervision from the larger teacher rather than only coarse-grained model-level signals.

---

### Note · Authors · 2026-01-06

I have read and agree with the venue's withdrawal policy on behalf of myself and my co-authors.